# Photocatalyzed regioselective hydrosilylation for the divergent synthesis of geminal and vicinal borosilanes

Degong Kong[1,2,5], Muliang Zhang [1,3,5], Yuchao Zhang[4], Zhenyang Yu[1], Hui Cao [1] ✉ & Jie Wu [1] ✉

Geminal and vicinal borosilanes are useful building blocks in synthetic chemistry and material science. Hydrosilylation/hydroborylation of unsaturated systems offer expedient access to these motifs. In contrast to the well-established transition-metal-catalyzed methods, radical approaches are rarely explored. Herein we report the synthesis of geminal borosilanes from α-selective hydrosilylation of alkenyl boronates via photoinduced hydrogen atom transfer (HAT) catalysis. Mechanistic studies implicate that the α-selectivity originates from a kinetically favored radical addition and an energetically favored HAT process. We further demonstrate selective synthesis of vicinal borosilanes through hydrosilylation of allyl boronates via 1,2-boron radical migration. These strategies exhibit broad scopes across primary, secondary, and tertiary silanes and various boron compounds. The synthetic utility is evidenced by access to multi-borosilanes in a diverse fashion and scaling up by continuous-flow synthesis.

Borosilanes, especially geminal and vicinal borosilanes, are valuable synthetic intermediates for the construction of complex molecules and functional materials[1,2]. The C–B and C–Si bonds in borosilanes can be chemoselectively converted into various functionalities through Suzuki-Miyaura coupling, alkylation, Zweifel olefination, oxidation, and conjugate addition[1,2]. The synthesis of borosilanes, mainly through transition-metal catalysis, has attracted much attention in the last decade (Fig. 1A)[3–18]. For instance, Morken and co-workers reported the synthesis of geminal borosilanes through Pt-catalyzed hydrosilylation of β-alkyl-substituted alkenyl boronates[8]. The regiocontrol is dictated by the formation of a stable α-boryl-organoplatinum intermediate after olefin insertion into the Pt-H bond. Hoveyda and co-workers developed selective formation of vicinal and geminal borosilanes from Cu-catalyzed hydroborylation of β-alkyl- and β-aryl-substituted alkenyl silanes, respectively[10]. Recently, Lu group disclosed elegant synthesis of borosilanes from

Co-catalyzed sequential hydrosilylation and hydroboration of arylacetylenes[13]. All four regioisomers could be selectively obtained through ligand control and dual catalysis relay strategy. Despite many progresses, the current methods usually require transition-metals and are frequently limited by the patterns of alkene/alkyne, silane and boron substrates. A widely applicable method for accessing geminal and vicinal borosilanes is missing.

In sharp contrast, synthesis of geminal or vicinal borosilanes via radical approaches is rarely explored[19–21]. Even though hydrosilylation of alkenes through hydrogen atom transfer catalysis or single electron transfer catalysis have been developed[19,22–28] and many alkenyl boronates are commercially available, regiocontrol in radical hydrosilylation of substituted alkenyl boronates remains challenging (Fig. 1A). The addition of radicals to alkenes relies on a complex interplay between polar, steric and enthalpic effects[29–31]. In terms of polarity, boronic ester is a weakly electron-withdrawing group, while

[1]Department of Chemistry, National University of Singapore, 3 Science Drive 3, Singapore 117543, Republic of Singapore. [2]School of Chemical Engineering & Technology, Harbin Institute of Technology, Harbin 150001, China. [3]Hefei National Laboratory for Physical Sciences at the Microscale and Department of Chemistry, University of Science and Technology of China, Hefei 230026, China. [4]Institute of Basic Medicine and Cancer (IBMC), Cancer Hospital of the University of Chinese Academy of Sciences, Hangzhou, Zhejiang 310022, China. [5]These authors contributed equally: Degong Kong, Muliang Zhang. ✉e-mail: charley5023@gmail.com; chmjie@nus.edu.sg

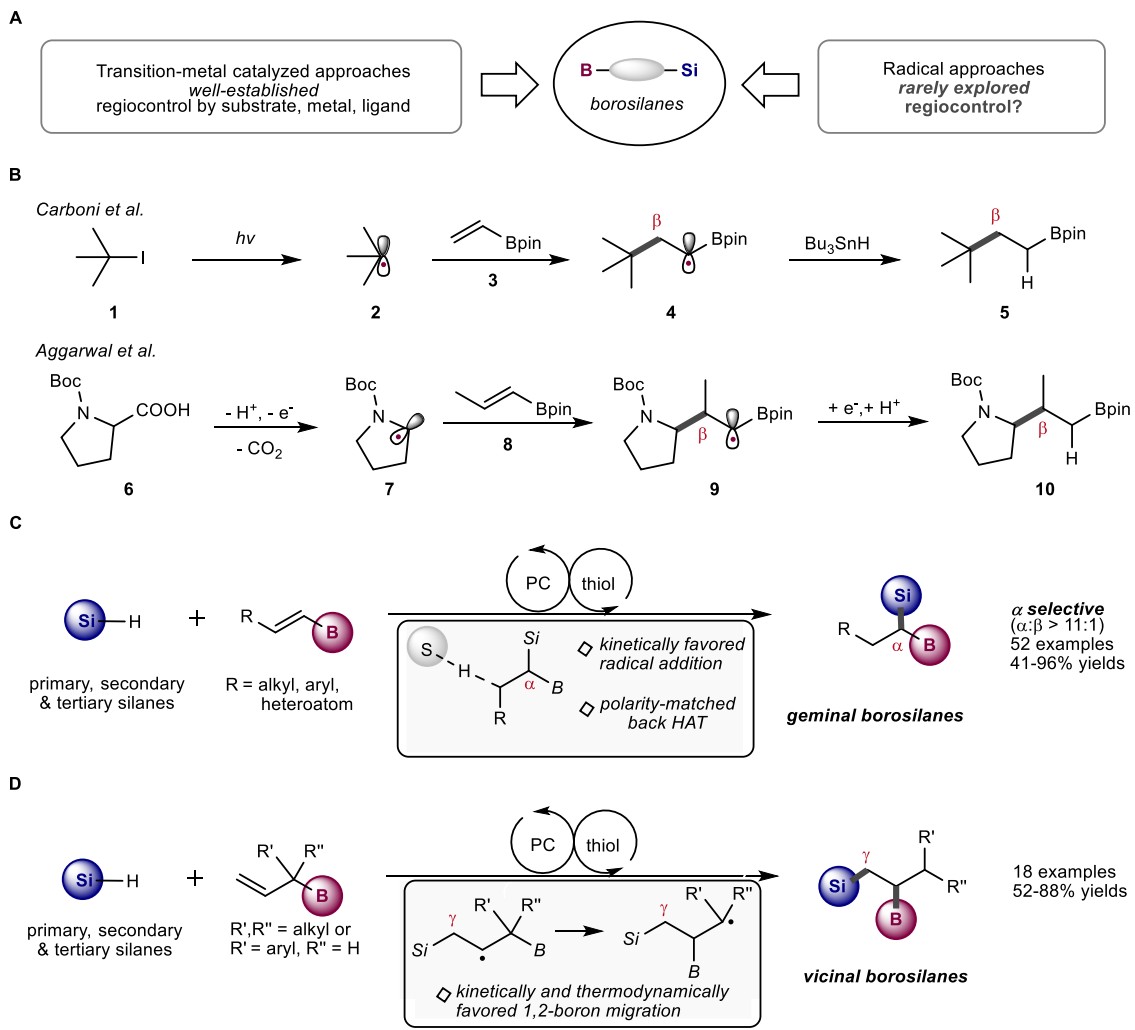

**Fig. 1 | Synthesis of geminal and vicinal borosilanes. A** Synthesis of borosilanes. **B** β-Selective alkylation of alkenyl boronates. **C** α-Selective silylation of alkenyl boronates (this work). **D** γ-Selective silylation of allyl boronates with concomitant 1,2-boron shift (this work).

silyl radicals could efficiently add to either electron-rich or electron-poor olefins[32]. Enthalpic regiocontrol, on the other hand, could be sought after. As silyl radical analogs, nucleophilic carbon-centered radicals were reported to add to alkenyl boronic esters in a β-selective fashion[33–38] because the formed α-boryl radicals (**4** and **9**, Fig. 1B) are stabilized by delocalization to the empty p-orbitals of the adjacent boron atoms[39–41]. The generated α-boryl radical intermediate could accept a hydrogen atom from tributyltin hydride or be reduced to α-boryl anion through single electron transfer (SET), furnishing β-alkylation products (**5** and **10**)[33,34].

In this work, geminal borosilanes are obtained from α-selective silylation of alkenyl boronic esters with photoinduced HAT catalysis[42,43] (Fig. 1C). The α-selectivity is general across a wide range of β-substituted alkenyl boronates (>11:1 selectivity in all cases). Furthermore, γ-selective silylation of allyl boronates is developed to provide access to vicinal borosilanes, relying on 1,2-boron radical migration (Fig. 1D). An extremely broad scope of borosilanes is effectively synthesized with these simple protocols, tolerating primary, secondary and tertiary silanes as well as different boryl alkenes. These metal-free transformations are operated under mild conditions and can be easily scaled-up in continuous flow reactors. Regiocontrollable stepwise hydrosilylation affords various types of multi-borosilanes which serve as attractive building blocks for material science.

## Results

### Development of α-selective silylation of alkenyl boronates

The reaction conditions for the synthesis of geminal borosilanes was first examined with *trans*−1-pentenylboronic acid pinacol ester as the model substrate (Supplementary Tables 1-3). The use of 1.2 equivalent of phenylsilane, 0.5 mol% photocatalyst 4-CzIPN (1,2,3,5-tetrakis-(carbazol-yl)−4,6-dicyanobenzene), 5 mol% HAT catalyst ethyl thioglycolate, and 5 mol% DIPEA (*N*,*N*-diisopropylethylamine) in methyl *tert*-butyl ether (0.1 M) under blue light irradiation was found optimal, affording the corresponding geminal borosilane in 92% yield and 14:1 regioselectivity. The choice of thiols has little effect on the regioselectivity (see Supplementary Table 1). Control experiments showed that the photocatalyst, light, and thiol are all necessary components for an effective transformation (Supplementary Table 3). The reaction proceeded much slower in the absence of DIPEA (92% vs. 54% yield after 4 h).

With the optimal conditions in hand, the substrate scope was evaluated (Fig. 2). A broad range of β-alkyl-substituted alkenyl boronates were incorporated to give geminal borosilanes **11**–**27** in 45-96% yields with high regioselectivity. Identical results were obtained regardless of whether *trans* or *cis* alkenyl boronates were used (**11**). Different functional groups including silyl ether (**16**), alkyl chloride (**17**), tetrahydropyranyl ether (**21**), and epoxide (**22**) were well-tolerated. Heterocycles such as proline (**24**), indole (**25**) and

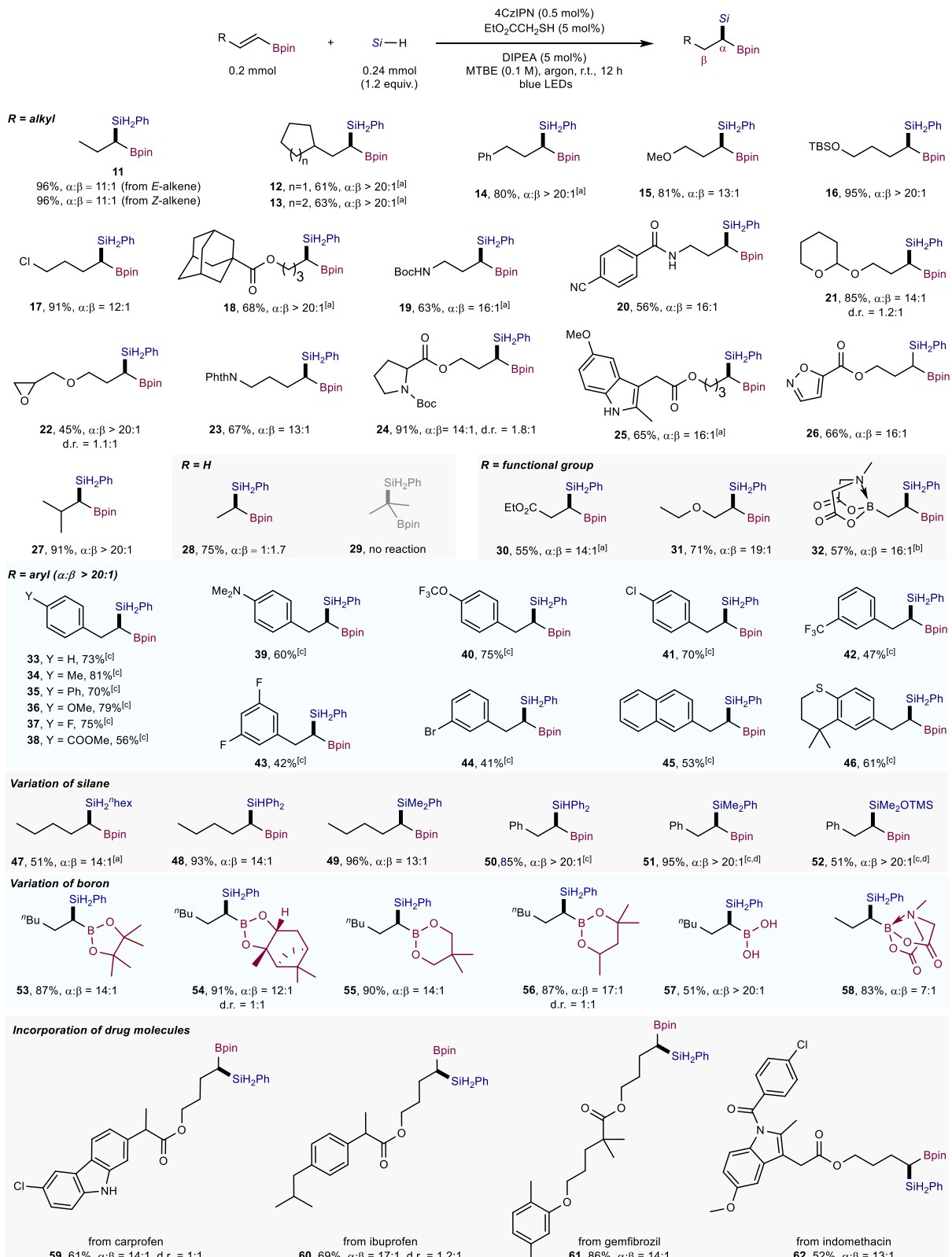

**Fig. 2 | Scope of synthesis of geminal borosilanes.** Reaction conditions: alkenyl boronate (0.2 mmol), silane (0.24 mmol), 4CzIPN (0.5 mol%), EtO$_2$CCH$_2$SH (5 mol%), DIPEA (5 mol%) in MTBE (methyl tert-butyl ether) (0.1 M) under irradiation with 40 W, 456 nm LED light at room temperature for 12 h under argon. Isolated yields. Regioselectivity was determined by GC analysis of the crude reaction mixture. [a]1 mol% 4CzIPN, 10 mol% EtO$_2$CCH$_2$SH and 10 mol% DIPEA. [b]With iPr$_3$SiSH in THF (0.1 M). [c]4CzIPN (1 + 1 mol%), EtO$_2$CCH$_2$SH (20 mol%), DIPEA (20 mol%) under irradiation with 80 W, 456 nm LED light for 48 h. [d]With iPr$_3$SiSH in MeCN (0.1 M).

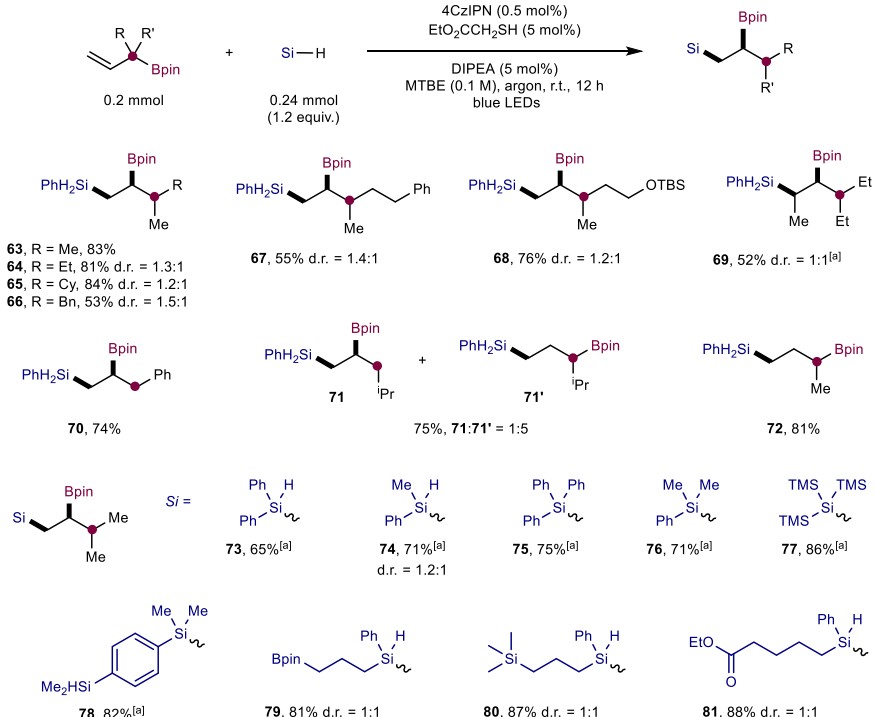

**Fig. 3 | Scope of synthesis of vicinal borosilanes.** Reaction conditions: allyl boronate (0.2 mmol), silane (0.24 mmol), 4CzIPN (0.5 mol%), EtO₂CCH₂SH (5 mol%), DIPEA (5 mol%) in MTBE (0.1 M) under irradiation with 40 W 456 nm LED light at room temperature for 12 h under argon. Isolated yields. [a]1 mol% 4CzIPN, 10 mol% iPr₃SiSH and 10 mol% DIPEA.

isoxazole (**26**) were accommodated as well. For unsubstituted vinyl boronate, a 1 to 1.7 ratio of α-silylation and β-silylation products (**28**) was noted. The reaction of isopropenylboronic acid pinacol ester delivered no hydrosilylation product (**29**) (Supplementary Table 5), demonstrating that β-silylation is indeed unfavorable under our reaction conditions. We continue to evaluate different substituents at the β position of alkenyl boronates such as an electron-withdrawing ester group (**30**), an electron-donating ethoxy group (**31**) and an electron-neutral boronic acid MIDA (*N*-methyliminodiacetic acid) ester group (**32**)[44]. The corresponding geminal borosilanes were all synthesized in good yields with high α-selectivity. Interestingly, (2-pinacolethenyl) boronate MIDA ester which contains sp²- and sp³-hybridized boron substituents on the olefin underwent selective silylation α to the Bpin moiety (**32**, 16:1). β-Aryl-substituted alkenyl boronates were then investigated, and the hydrosilylation products were formed with exclusive α-selectivity (**33–46**, 41–81% yields), regardless of electron-withdrawing or electron-donating groups on the aryl substituent. Some sensitive functionalities in transition-metal catalyzed hydrosilylation or hydroboration, including aryl chlorides (**41**) and aryl bromides (**44**) were well-tolerated with our protocol.

The incorporation of different patterns of silanes such as primary, secondary and tertiary silanes in hydrosilylation could lead to silicon products with distinct reactivities[45]. However, transition-metal catalyzed methods for borosilane synthesis are generally limited to one type of silanes (e.g. secondary silanes)[7–14]. It was found that primary, secondary and tertiary silanes were all competent substrates in our transformation, delivering structurally diverse geminal borosilanes effectively (**47–52**). A siloxane substrate was converted to **52** in 51% yield, indicating the potential utility of this method in silicone polymer chemistry[46]. Moreover, diverse boryl alkenes reacted smoothly under the optimal reaction conditions, embracing different boronates (**53–56**), a free boronic acid (**57**), and a boronic acid MIDA ester (**58**). The good efficiency, regioselectivity and functional-group tolerance prompted us to evaluate the potential of this method for derivatization

of complex drug-like molecules, and the reactions proceeded well with carprofen, ibuprofen, gemfibrozil and indomethacin derivatives (**59–62**).

## Development of γ-selective silylation of allyl boronates

Having established a general method to access geminal borosilanes, we next turned our attention to vicinal borosilanes. The extremely low reactivity of isopropenylboronic acid pinacol ester (**29**) suggests that a completely different strategy is required to synthesize vicinal borosilanes. We were inspired by the studies on 1,2-boron radical migration[47–51] and found that the switch of alkene substrates to α-substituted allyl boronates readily delivered vicinal borosilanes (Fig. 3). This strategy worked well for α,α-dialkyl-substituted allyl boronates (**63–69**, 52–84% yields), and the presence of a methyl group at the γ position was tolerated (**69**). α-Aryl-substituted allyl boronates were also competent substrates (**70**, 74%). Changing the α-substituent of an allyl boronate to an isopropyl group furnished the migrated product **71** and non-migrated product **71'** in 1:5 ratio, while α-methyl allyl boronate only gave non-migrated product **72**. The trends are in good accordance with the decreased stability of the radical intermediate derived from 1,2-boron radical migration[47–51]. Likewise, a wide range of primary, secondary and tertiary silanes participated in the silylation/migration cascade smoothly (**73–81**, 65–88% yields). The formed vicinal borosilanes **78–81** contain additional handles such as silane, boronate and ester groups for further molecular diversification.

## Further synthetic utilizations

The synthetic utility of the divergent photo-HAT hydrosilylation is further illustrated in Fig. 4. The selective synthesis of geminal borosilane **82** was achieved by two metal-free transformations from the herbicide clodinafop-propargyl in one pot (65% yield, Fig. 4a). The highly selective monofunctionalization of trihydrosilanes and dihydrosilanes through photo-HAT catalysis offers opportunities for stepwise decoration of silicon atoms to access structurally diverse multi-

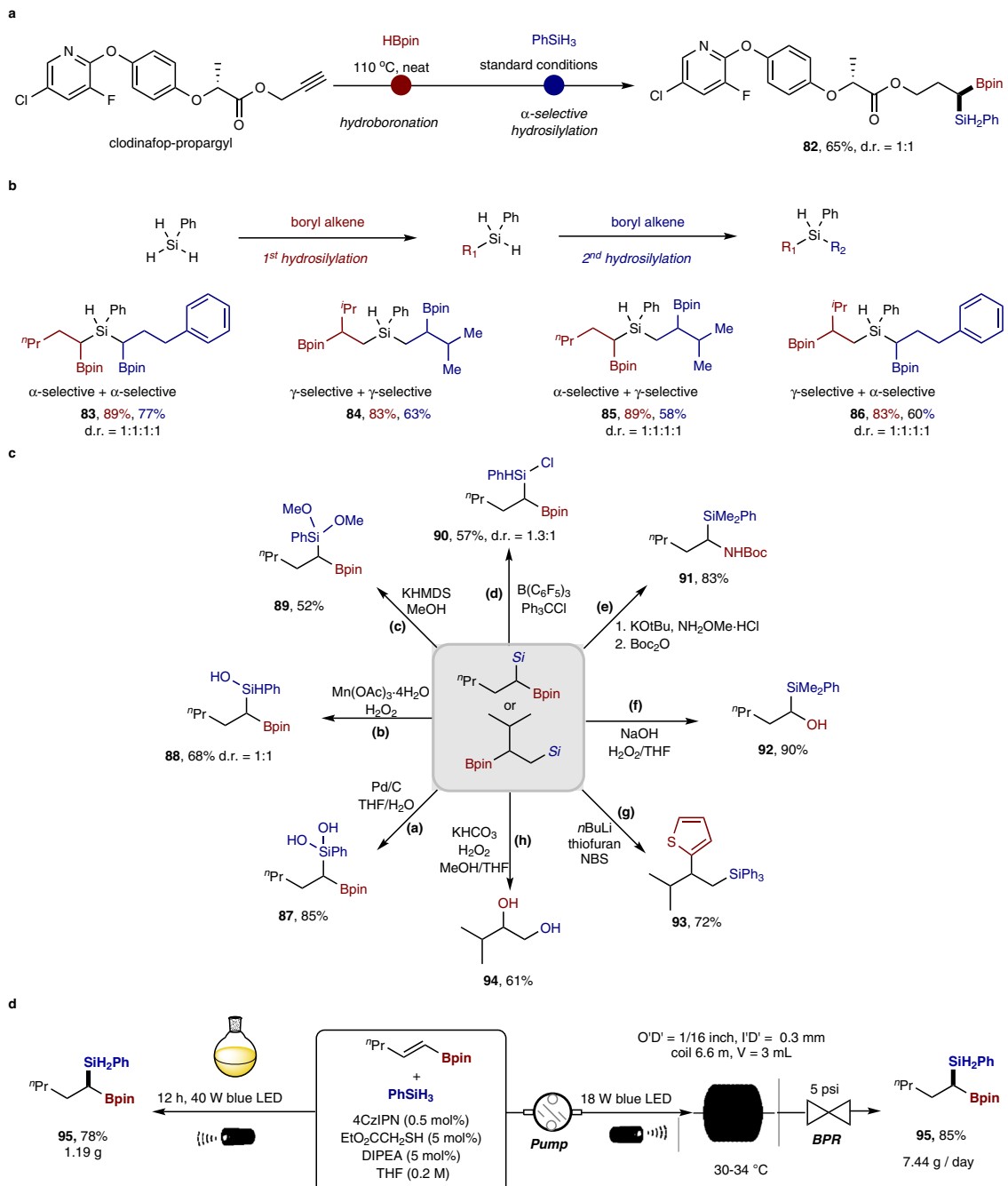

**Fig. 4 | Further synthetic utilizations. a** One-pot silylboration of clodinafop-propargyl. **b** Synthesis of multi-borosilanes from PhSiH₃. **c** Derivatization of silylboronates. **d** Gram-scale synthesis in batch and continuous-flow reactors.

borosilanes. By selectively sequencing the α-selective silylation of alkenyl boronate or γ-selective silylation of allyl boronate processes, different types of multi-borosilanes (**83**–**86**) could be obtained (Fig. 4b). The boryl or silyl groups in the geminal and vicinal borosilanes are poised for conversion to a range of valuable products. For example, the silyl group in the products could be converted through Si-H bond functionalization to produce silicon-containing compounds such as silanols (**87**, **88**), silylether (**89**), silylchloride (**90**), as shown in Fig. 4c. The boryl group represents an extremely versatile synthetic handle in organic synthesis, which are known to undergo amination, oxidation and arylation to assemble C–N, C–O and C–C bonds, respectively. To showcase this flexibility, the boryl group in the products was converted to amines, alcohols, and arenes with the silyl

group intact (**91**-**93**) (Fig. 4c). By tuning the oxidative conditions, both the silyl and boryl group underwent oxidation, giving rise to 1,2-diol product **94**. Furthermore, gram-scale reactions were demonstrated in a batch reactor (**47**, 78% yield, 1.19 g) as well as an operationally simple continuous-flow reactor (**47**, 85%, 7.44 g per day production) (Fig. 4d).

## Mechanistic considerations
To shed some light on the reaction mechanism and the origin of regioselectivity, a series of control experiments as well as computational studies were conducted. The radical nature of these transformations was confirmed by radical scavenger and radical clock experiments (see Supplementary Discussion). Stern–Volmer fluorescence quenching studies indicated that the excited photocatalyst can

**Fig. 5 | Proposed mechanism of α-selective silylation of alkenyl boronates.** A plausible reaction mechanism for the hydrosilylation of alkenyl boronates proposed based on all experimental results and previous literature evidence. PC photocatalyst.

be reductively quenched by the mixture of thiol and DIPEA (Supplementary Fig. 8). A plausible reaction mechanism for the hydrosilylation of alkenyl boronates was proposed (Fig. 5)[22,23]. The excited 4CzIPN oxidizes the thiol in the presence of DIPEA. Cyclic voltammetry measurements (Supplementary Fig. 7) and DFT calculations suggest the formation of a thiyl radical species **I**. The thiyl radical **I** abstracts a hydrogen atom from silane to afford the silyl radical and thiol **N**. Computational studies suggested that the complexation with DIPEA stabilizes intermediate **I** and facilitates hydrogen atom abstraction from silane (Supplementary Figs. 20, 21). Subsequently, the silyl radical adds to the α-position of the alkenyl boronate to deliver an alkyl radical intermediate which undergoes polarity-matched HAT process with thiol **N** to give the geminal borosilane product. The HAT steps in the proposed mechanism are supported by deuterium-labeling experiments (Supplementary Figs. 9–14). At this stage, the thiyl radical **I** could oxidize the reduced photocatalyst 4CzIPN•⁻ to close the photocatalytic cycle. Alternatively, thiyl radical **I** could react further with silane via radical chain pathways. However, the light on/off experiments and the calculated quantum yields ($\Phi = 0.109$) did not support an efficient radical chain process (Supplementary Figs. 15–17). Computational analysis was then conducted. It was found that the back electron transfer from the reduced photocatalyst 4CzIPN•⁻ to thiyl radical **I** has an energy barrier of only 1.02 kcal/mol, while HAT from silane to thiyl radical **I** has a much higher energy barrier of 7.63 kcal/mol (Supplementary Figs. 20 and 21). We attribute the ineffective chain propagation and low quantum yield to the unproductive back electron transfer.

We next sought to elucidate the origin of α-selectivity in the hydrosilylation of alkenyl boronates. It is noted that α-selectivity was also observed in the hydrosilylation of alkenyl boronates using engineered carbon nitrides as heterogeneous photocatalysts[20]. However, the reaction scope is very limited and the reason for α-selectivity in this heterogeneous process remains unknown. Here, the silyl radical addition and subsequent hydrogen atom transfer with thiols were analyzed by calculations using $(E)$−1-pentenylboronic acid pinacol ester (**R¹**) and phenylsilane as the model substates (Fig. 6 and Supplementary Fig. 23). The calculated energy diagram illustrates that the addition of silyl radical to alkenyl boronic esters determines the regioselectivity because the

transition states (**S¹** or **S²**) have the highest energy in the reaction pathways[52,53]. This also explains why similar regioselectivity was observed with different thiols (Supplementary Table 1). The energy barrier of silyl radical adding to α-position of **R¹** is 1.64 kcal·mol⁻¹ lower than that to β-position (**S¹** vs **S²**), which means the α-addition rate is approximately 16 times faster than β-addition. This is very close to the observed selectivity in the crude reaction mixture (α/β = 14:1). Despite higher stability of the generated intermediate **T²** after β-addition[33–39], there are two reasons for the kinetic-controlled α-selectivity. The radical addition processes are nearly irreversible at room temperature, thus the equilibrium between α- and β-addition products cannot be reached. Moreover, HAT from thiol **N** to the radical intermediate **T¹** is both kinetically and thermodymically favored ($\Delta G^{\neq} = 11.71$ kcal·mol⁻¹, $\Delta G = -7.89$ kcal·mol⁻¹) due to polarity-match[29,54]. The higher HAT rate of **T¹** compared to **T²** further reduces the concentration of the radical **T¹**. Overall, the kinetically favored radical addition and energetically favored back HAT process contribute to the α-selective silylation of alkenyl boronates. Similar elucidation is also found for *cis*-alkenyl boronates (Supplementary Fig. 23).

A plausible mechanism for the hydrosilylation of allyl boronates was also proposed (Supplementary Fig. 19), and the γ-selectivity was analyzed by calculations (Fig. 7). The silyl radical **M** generated from HAT selectively adds to the sterically more accessible γ-position of the allyl boronic ester to give a β-boryl carbon-centered radical intermediate **T⁵**. The α-addition is not favored (Supplementary Fig. 24). At this stage, a 1,2-boron migration process influenced by the α-substituents on the allyl boronates took place[47–50]. The migration was steered by thermodynamic effects to generate a more stable carbon radical **T⁷** which undergoes polarity-matched HAT process with thiol **N** to give vicinal borosilanes. DFT calculations indicate the migration energy barrier for α,α-dimethyl allyl boronate is low ($\Delta G^{\neq} = 9.23$ kcal·mol⁻¹) and the rearranged radical intermediate **T⁷** is more stable than the non-migrated radical **T⁵** ($\Delta G = -1.69$ kcal·mol⁻¹) (Supplementary Fig. 24). Moreover, the HAT reaction rate of rearranged radical **T⁷** with thiol is much faster compared to **T⁵**, thereby allowing selective synthesis of vicinal borosilanes. Finally, the generated thiyl radical **I** could accept an electron from 4CzIPN•⁻ to close both catalytic cycles or engage in chain propagations.

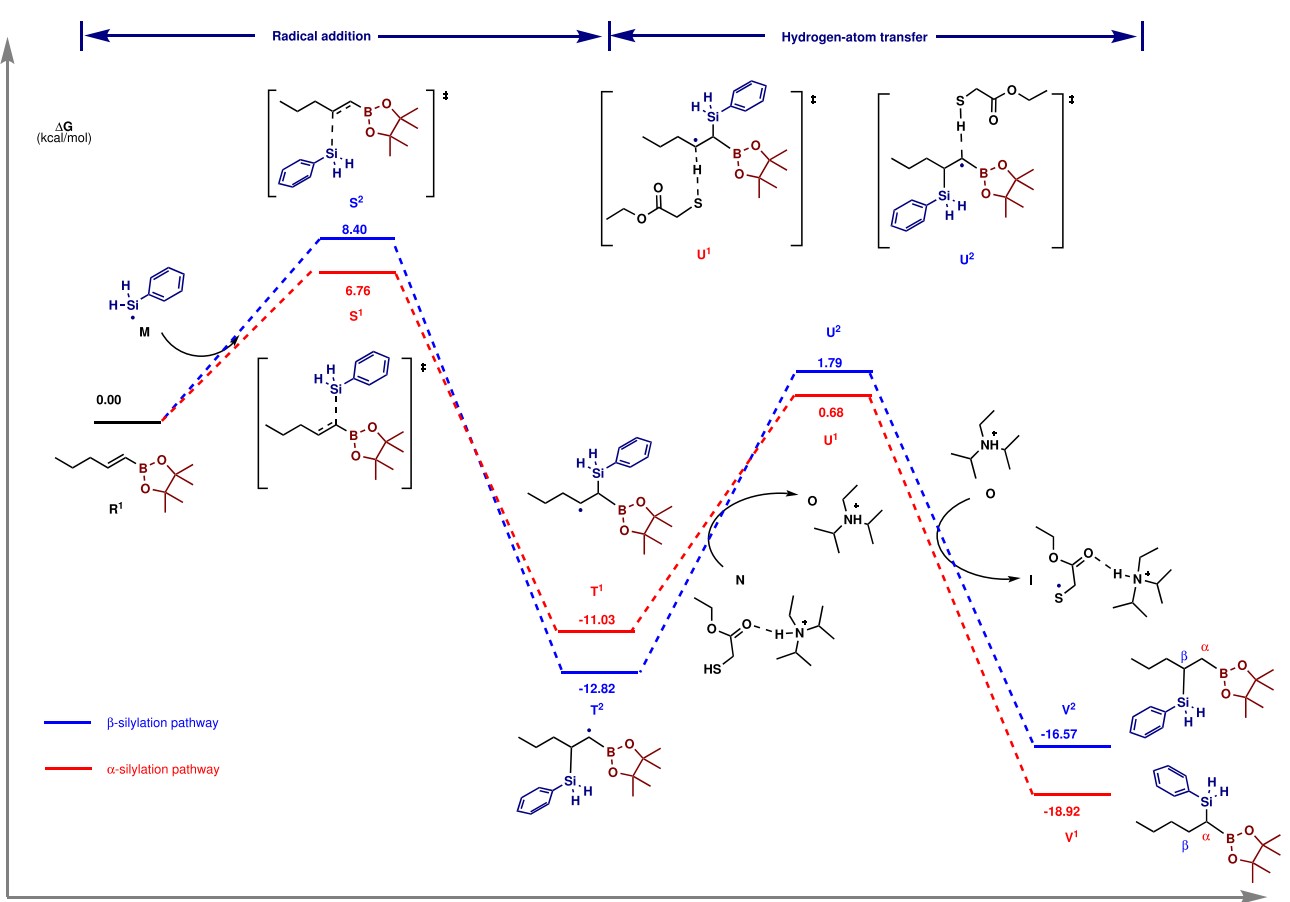

**Fig. 6 | Free energy diagram for the hydrosilylation of (*E*)-1-pentenylboronic acid pinacol ester.** Geometry and energy were calculated at M06-2X-D3/6-31 G** (IEFPCM, eps = 2.6, epsinf = 1.874) and M06-2X-D3/may-cc-pVTZ (IEFPCM, eps = 2.6, epsinf = 1.874) level of theory respectively.

In summary, we have developed efficient photo-HAT catalyzed α-selective silylation of alkenyl boronates and γ-selective silylation of allyl boronates, providing a broad range of structurally diverse geminal and vicinal borosilanes. Unlike transition-metal catalysis, the regioselectivity is determined by kinetic and thermodynamic effects in radical addition and HAT processes. These protocols featured merits such as metal-free, atom-economy, extremely broad substrate scopes and good functional group tolerance. The silyl or boryl groups in the products demonstrated versatile synthetic utility for subsequent diversification. Gram-scale reactions were smoothly achieved in a batch reactor and an operationally simple continuous-flow reactor.

## Methods

### General procedure of α-selective silylation of alkenyl boronates to synthesized β-alkyl geminal borosilanes

A 10 mL microwave tube equipped with a magnetic stir bar was charged with 4CzIPN (0.8 mg, 0.001 mmol, 0.5 mol%), alkenyl boronate (0.2 mmol), silane (0.24 mmol, 1.2 equiv.) and anhydrous MTBE (2 mL). The tube was capped with a Supelco aluminum crimp seal with septum (PTFE/butyl). The resulting mixture was cooled to 0 °C using an ice-water bath and bubbled with an argon balloon for 10 min. DIPEA (1.8 μL, 0.01 mmol, 5 mol%) and EtO$_2$CCH$_2$SH (1.1 μL, 0.01 mmol, 5 mol%) were then added. After that, the reactor was placed under a blue LED (Kessil light, 40 W, 456 nm) and irradiated for 12 h at room temperature. The solvent was removed under vacuum. Purification by flash column chromatography on silica gel (eluent: *n*-hexane/EtOAc mixtures) gave the desired product.

### General procedure of α-selective silylation of alkenyl boronates to synthesized β-aryl geminal borosilanes

A 10 mL microwave tube equipped with a magnetic stir bar was charged with 4CzIPN (1.6 mg, 0.002 mmol, 1 mol%), alkenyl boronate (0.2 mmol), silane (0.24 mmol, 1.2 equiv.) and anhydrous MTBE (2 mL). The tube was capped with a Supelco aluminum crimp seal with septum (PTFE/butyl). The resulting mixture was cooled to 0 °C using an ice-water bath and bubbled with an argon balloon for 10 min. DIPEA (7.2 μL, 0.04 mmol, 20 mol%), and EtO$_2$CCH$_2$SH (4.4 μL, 0.04 mmol, 20 mol%) were then added. After that, the reactor was placed under blue LED (Kessil light, 80 W, 456 nm) and irradiated for 24 h at room temperature. And then, add 4CzIPN (1.6 mg, 0.002 mmol) into the microwave tube in the glovebox, and removed it from the dry box. The reaction was irradiated for additional 24 h under the same conditions. The solvent was removed under vacuum. Purification by flash column chromatography over silica gel (eluent: *n*-hexane/EtOAc mixtures) gave the desired product.

### General procedure of γ-selective silylation of allyl boronates to synthesized vicinal borosilanes

A 10 mL microwave tube equipped with a magnetic stir bar was charged with 4CzIPN (0.8 mg, 0.001 mmol, 0.5 mol%), allyl boronate (0.2 mmol), silane (0.24 mmol, 1.2 equiv.) and anhydrous MTBE (2 mL). The tube was capped with a Supelco aluminum crimp seal with septum (PTFE/butyl). The resulting mixture was cooled to 0 °C using an ice-water bath and bubbled with an argon balloon for 10 min. DIPEA (1.8 μL, 0.01 mmol, 5 mol%) and EtO$_2$CCH$_2$SH (1.1 μL, 0.01 mmol, 5 mol%) were then added. After that, the reactor was placed under a blue LED (Kessil light, 40 W, 456 nm) and irradiated for 12 h at room

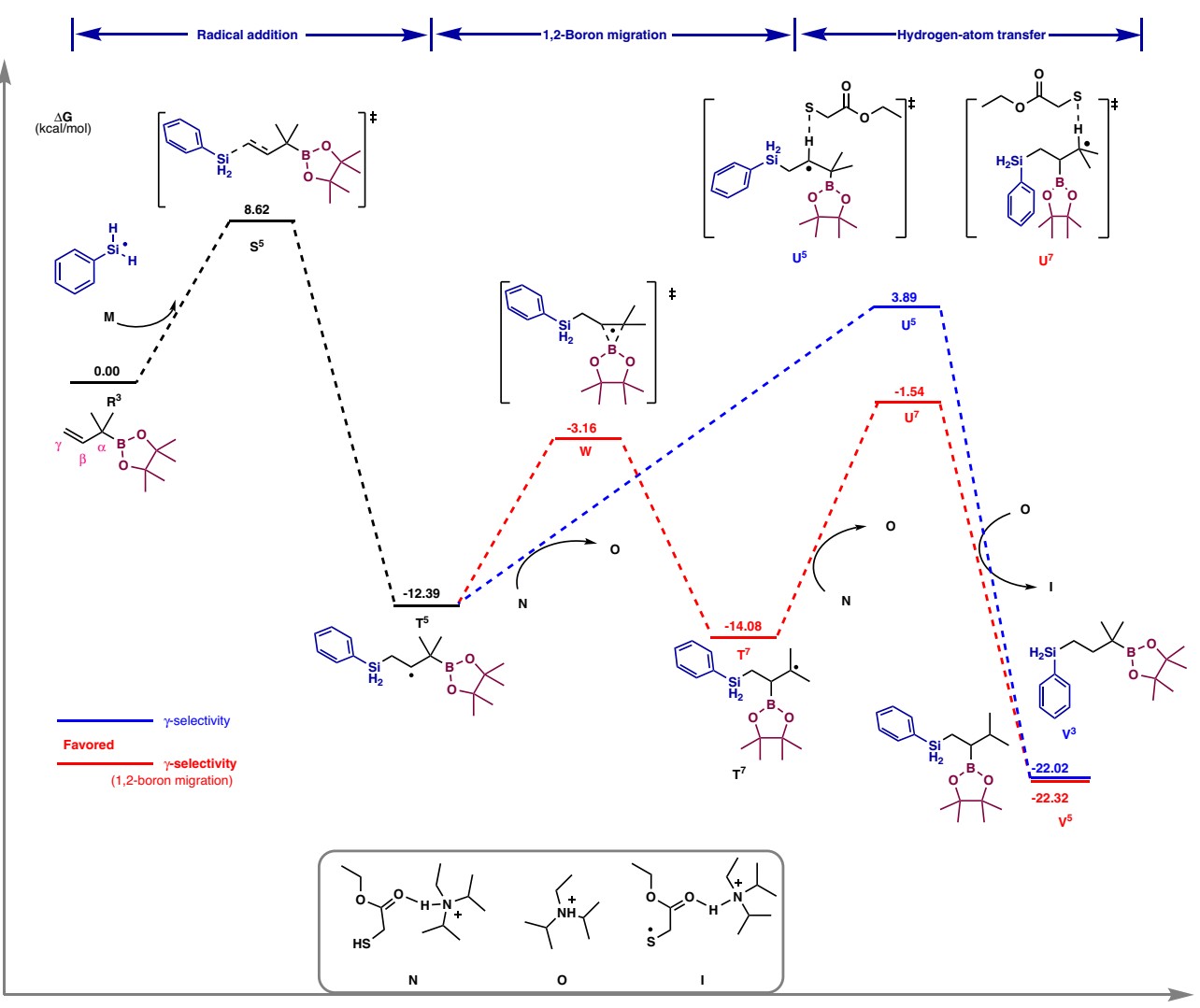

**Fig. 7 | Free energy diagram for the hydrosilylation of 1,1-dimethyl-2-propen–1-boronic acid pinacol ester.** Geometry and energy were calculated at M06-2X-D3/6-31 G** (IEFPCM, eps = 2.6, epsinf = 1.874) and M06-2X-D3/may-cc-pVTZ (IEFPCM, eps = 2.6, epsinf = 1.874) level of theory respectively.

temperature. The solvent was removed under vacuum. Purification by flash column chromatography over silica gel (eluent: *n*-hexane/EtOAc mixtures) gave the desired product.

## Data availability
The authors declare that all other data supporting the findings of this study are available within the article and Supplementary Information files, and also are available from the corresponding author upon request. Source data are provided with the publication. Source data are provided with this paper.

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

## Acknowledgements

We gratefully acknowledge the financial support provided by the Ministry of Education (MOE) of Singapore (MOET2EP10120-0014, J.W.), National Natural Science Foundation of China (Grant No. 22071170, J.W.), and NUS Chongqing Research Institute.

## Author contributions

D.K., M.Z., H.C. and J.W. conceived and designed the investigations. Y.Z. conducted DFT calculations. D.K., M.Z., H.C. and Z.Y. performed the experiments. H.C. and J.W. wrote the manuscript.

## Competing interests

The authors declare no competing interests.
