## [Peer Review File · Nature Communications]

Photocatalyzed Regioselective Hydrosilylation for the Divergent Synthesis of Geminal and Vicinal BorosilanesReviewers' Comments:

Reviewer #1:

Remarks to the Author:

This manuscript from Cao and Wu and co-workers describes the divergent hydrosilylation of vinyl boron to access either geminal or vicinal borylated/silylated carbon center.

The methodology is efficient and the products were obtained in good to excellent yields and the regioselectivity is very good. In addition, the scopes of these transformations are very good and demonstrate the applicability of the method. This is an interesting and straightforward methodology to build up borylated/silylated carbon-centered containing molecules. The authors delineated a plausible mechanism supported by experiments and DFT calculations.

This work is a nice piece of work and deserves publication. However, I have several concerns with this manuscript and particularly regarding its relevance to the readership of Nature Communications:

1/ The concept of this reaction was already disclosed by the authors (ACIE, 2020, 4009) and this work is rather an extension (under slightly modified conditions) to vinyl boron derivatives. Therefore, the novelty and originality of this work do not meet the standard required by the journal.

2/ The literature is not well cited, several key papers on hydrosilylation based on a SET for the silyl radical by photoredox or electrochemical process are missing. This is important to include these references for the sake of the readership, to understand the novelty of this work.

3/ In the SI, postfunctionalization products are not characterized; Some C are missing in the product description, probably the C-B center. If so, it has to be mentioned. All IR analyses are missing, as well as R_f value. When diastereoisomeric mixtures were obtained, identification of the major and minor diastereoisomers is needed on the most significant signals.

As minor comments:

4/ The mechanistic study should be fully included in the manuscript for the sake of the readership.

5/ regarding compounds description: 54: the dr is not mentioned; 24 & 60 the stereogenic center of proline and ibuprofen is not mentioned.

6/ the discussion section is a conclusion.

Overall, this paper is interesting for the community. However, I am skeptical about the relevance of this work for publication in a high-standard journal, i.e. Nature Communications in terms of novelty and urgency. Of course, this manuscript deserves publication in a more specialized journal focusing on either organic chemistry or catalysis.

Reviewer #2:

Remarks to the Author:

The manuscript by Cao, Wu, and co-workers describes a range of photoredox-catalysed hydrosilylations of alkene-containing boronic esters to synthesise borosilane products. This work builds on the authors' previously reported photoredox/HAT-catalysed alkene hydrosilylations (reference 23) by extending the reactions to alkenyl and allyl boronic esters. In contrast to their earlier work, the results for alkenyl boronic esters show unexpected regioselectivity, with the silyl radicals adding alpha to the boron centre to give geminal borosilanes. In addition, for allyl boronic esters, silyl radical addition is followed by a 1,2-boron shift to give vicinal borosilanes, thus demonstrating the versatility of the reaction conditions. Although they have reported the same regioselective hydrosilylations of alkenyl boronic esters previously with a different photoredox catalyst (reference 20), the present work provides a significantly more detailed study. The substrate scope is extensive for both the alkenyl boronic esters and the silanes, with high yields and regioselectivities obtained in most cases. The scope and limitations of the allyl boronic esters is also well described. The sequential reactions and derivatizations shown in Figure 2 are a nice demonstration of the synthetic utility of the chemistry, as are the scale up reactions. Finally, the authors present a detailed mechanistic study of the catalytic cycle and the regioselectivity, with DFT calculations providing a convincing explanation for the observed selectivity.

Based on the interesting regioselectivity and synthetic utility of this chemistry, I recommend publication in Nature Communications after addressing the following points.

Catalytic cycle:

- In Figure 3, the arrow showing the direct transformation of N to C in the mechanism is confusing, since it should proceed via HAT and then SET rather than simple deprotonation. This should be modified (or removed like in Supplementary Figure 18).
- In Figure 3, "PCH•-" should be "PC•-" because the proton remains on I.
- "Both light on/off experiments and the calculated quantum yields ($\Phi = 0.109$) supported photocatalytic cycles instead of radical chain propagation." – The mechanism in Fig. 3 is a radical chain because N and I are interconverted by sequential HAT reactions. The mechanism should be redrawn to make this clearer. As drawn, the photocatalytic cycle should be non-productive because C is converted to I and then I back to C (back electron transfer), therefore hydrosilylation should only proceed if some of I can initiate a chain.
- Based on the energies in Supplementary figure 20 the catalytic cycle looks energetically disfavored. Mainly because intermediates G+I undergo back electron transfer with a barrier of 1.02 kcal/mol, whereas HAT with the silane has a much higher barrier of 7.63 kcal/mol. This should be discussed. Perhaps it indicates that a radical chain propagation must be occurring to outcompete unproductive back electron transfer.
- The competing unproductive back electron transfer could also account for the low quantum yield because it would result in an inefficient initiation process.

DIPEA/thiol complex:

- "1H NMR spectroscopy showed the formation of complex C (Fig. 3) upon mixing ethyl thioglycolate (A) with equimolar DIPEA (B) at room temperature (Supplementary Fig. 7)." – The NMR studies for the formation of a DIPEA/thiol complex are not convincing. The chemical shifts have not changed, only the exchangeable proton on the thiol is no longer visible, therefore this is not proof for complexation. Complex formation should result in changes in the chemical shifts of the protons on both DIPEA and the thiol. Furthermore, the DFT results in Supplementary Figure 20 do not suggest that complexation should be seen by NMR, since it is 3.5 kcal/mol higher in energy.
- The NMR studies should be performed in a solvent that is relevant to the reaction, such as THF-d8. Alternatively, an entry for CHCl₃ (or CDCl₃) should be added to Supplementary Table 2 to show that the chemistry still works in this solvent.
- "Stern–Volmer fluorescence quenching and cyclic voltammetry studies indicated that the excited photocatalyst was reductively quenched by the thiol-DIPEA complex C" – Quenching studies show the plot of DIPEA quenching has a larger gradient (87.7) than that of the DIPEA/thiol mixture (81.3), implying that DIPEA is the predominant reductive quencher. This should be accounted for in the mechanistic discussion.
- Cyclic voltammetry could potentially be used to confirm the formation of the DIPEA/thiol complex. However, this would require separate CVs for the thiol, DIPEA, and the thiol/DIPEA mixture. Also, these should be performed in MTBE, since the reaction fails in MeCN.
- "The conformation of thiyl radical I and thiol N was suggested by DFT calculations (Supplementary Figs. 20 and 21)" – what is the relevance of this? It is not clear from the discussion in the Supplementary Information.
- Significant emphasis has been placed on the importance of the complex between the thiol and DIPEA, despite the reaction proceeding with only slightly reduced efficiency in the absence of DIPEA. If sufficient evidence for its formation cannot be provided, the mechanism and discussion should be modified accordingly.

Other points:

- "For unsubstituted vinyl boronate, a 1 to 1.7 ratio of α -silylation and β -silylation products (28) was noted, possibly due to the steric effect during the radical addition" – could this also be a result of a 1,2-boron shift of the primary alkyl radical intermediate?
- Page S37 discusses how to "determine the photon flux of the spectrophotometer". This should be the

photon flux of the Kessil lamp used to measure the quantum yield.

- The method for determining the photon flux states 3.0 mL of ferrioxalate solution was used. But later it states $V_1 = 2.0$ mL and 2.0 mL of MTBE was used in the reaction to measure the quantum yield. These numbers need to be checked and corrected.
- In Supplementary Figure 19, the gamma symbols have been replaced by "?".

Signed: Adam Noble

Reviewer #3:

Remarks to the Author:

The authors in this manuscript report synthesis of geminal and vicinal borosilanes via photocatalyzed regioselective hydrosilylation. The reported synthetic protocol looks interesting. Detailed mechanistic studies were also performed. Since I am not a synthetic chemist, I will let colleagues in the synthetic area comment on the importance of the reactions reported. As requested by the editor, I will mainly comment on the DFT calculation part.

(1) The DFT results are clearly consistent with the experimental findings regarding the regioselectivity. While the authors employed the steric argument to rationalize γ -selective silylation of allyl boronates, it is unclear why the steric effect becomes unimportant in α -selective silylation of alkenyl boronates. The authors owe readers a clear picture regarding what they have said on the selectivity issues. In the silylation of alkenyl boronates, radical addition intermediate T2 (Figure 4) is more stable than T1. However, the barrier leading to T2 is higher. An explanation is needed for readers to understand the DFT results better.

(2) Since this is a manuscript submitted to Nature Communications, I expected that the authors should explain why they use M06-2X functional instead of others in their calculations. I think the authors should also use some other functionals to examine if different functionals give consistent results.

(3) The DFT results provided in the Supplementary Information are poorly organized and described.

(i) First, I expect that the energy profiles given in Fig. S20, 22 and 23 should have the same quality as the one presented in Fig. 4 in the main text. Readers easily comprehend the energy profiles given in the main text Fig. 4. However, it is very hard to understand and comprehend the energy profiles given in the SI.

(ii) The description and discussion given in the SI are also far poorer than those given in the main text. The English should be polished with significant effort.

(iii) Fig. S20 should be completely redrawn. I do not understand the symbol placed between G+M+N and G+I. In the related discussion, the authors should clearly spell out the most favorable pathway. Ideally, the authors provide a sketch of the favorable cycle in the SI.

Response to Reviewer # 1

Comment 1: *This manuscript from Cao and Wu and co-workers describes the divergent hydrosilylation of vinyl boron to access either geminal or vicinal borylated/silylated carbon center. The methodology is efficient and the products were obtained in good to excellent yields and the regioselectivity is very good. In addition, the scopes of these transformations are very good and demonstrate the applicability of the method. This is an interesting and straightforward methodology to build up borylated/silylated carbon-centered containing molecules. The authors delineated a plausible mechanism supported by experiments and DFT calculations. This work is a nice piece of work and deserves publication. However, I have several concerns with this manuscript and particularly regarding its relevance to the readership of Nature Communications:*

Response: We sincerely thank Reviewer #1 for the comments and valuable suggestions.

Comment 2: *1) The concept of this reaction was already disclosed by the authors (ACIE, 2020, 4009) and this work is rather an extension (under slightly modified conditions) to vinyl boron derivatives. Therefore, the novelty and originality of this work do not meet the standard required by the journal.*

Response: Thanks for pointing out this concern. The hydrosilylation of alkenes through photoinduced hydrogen atom transfer (HAT) catalysis has been disclosed by our group before (*Angew. Chem. Int. Ed.* **56**, 16621–16625 (2017)). However, the major novelty of this work is the regiocontrol in radical hydrosilylation and mechanistic insight into the unusual regioselectivity. We would like to quote some of the comments from reviewer 2 as follow: “*This work builds on the authors’ previously reported photoredox/HAT-catalysed alkene hydrosilylations by extending the reactions to alkenyl and allyl boronic esters. In contrast to their earlier work, the results for alkenyl boronic esters show unexpected regioselectivity, with the silyl radicals adding alpha to the boron centre to give geminal borosilanes. In addition, for allyl boronic esters, silyl radical addition is followed by a 1,2-boron shift to give vicinal borosilanes, thus demonstrating the versatility of the reaction conditions. The authors present a detailed mechanistic study of the catalytic cycle and the regioselectivity, with DFT calculations providing a convincing explanation for the observed selectivity.*” In fact, compared to well-studied transition-metal catalysis, regiocontrol in radical hydrofunctionalization processes is much less explored. For example, in most studies involving the addition of a radical to an alkene (Giese process), biased alkenes such as activated alkenes (e.g., styrenes) or terminal alkenes are used, relying on strong enthalpic or steric preference. There are very few studies on relatively unbiased alkenes. In this study, we show the unusual α -selectivity in hydrosilylation of substituted alkenyl boronates originates from kinetic preference during the radical addition process and enthalpic factors during the back HAT process. Similarly, the interplay of steric and enthalpic effects contributes to γ -selective silylation of allyl boronates. This study provides evidence that radical approaches could be used for more challenging regiocontrol, and will help pushing the boundaries of selective radical functionalization of alkenes.

Other important characters of this work include: 1) a rare example of α -selective radical functionalization of alkenyl boronates because the addition of radicals to alkenyl boronates has been known to be β -selective; 2) the first general strategy to synthesize geminal and vicinal borosilanes via radical pathways; 3) green and sustainable synthesis of borosilanes; 4) access to diverse multi-borosilanes which serve as attractive building blocks for material science.

Comment 3: *2) The literature is not well cited, several key papers on hydrosilylation based on a SET for the*

3 Science Drive 3, Singapore 117543

Website: www.wujiegrouppnus.com

silyl radical by photoredox or electrochemical process are missing. This is important to include these references for the sake of the readership, to understand the novelty of this work.

Response: Thanks for this valuable suggestion. We have added the relevant references (ref. 19, 22-26).

Comment 4: 3) *In the SI, postfunctionalization products are not characterized; Some C are missing in the product description, probably the C-B center. If so, it has to be mentioned. All IR analyses are missing, as well as R_f value. When diastereoisomeric mixtures were obtained, identification of the major and minor diastereoisomers is needed on the most significant signals.*

Response: Thanks for this important suggestion. We have added these characterization data to the Supplementary Information. The missing C in ¹³C NMR because of the nearby boron is mentioned. The IR analyses and R_f values have been added. The diastereomer ratios of products **83, 84, 86** have been added. We have tried to identify the major diastereomer in compounds **21, 22, 24, 60, 64-68, 74** and **90** using 2D NMR, but no useful information was obtained. In some cases, the stereocenters are far away from each other. In other cases, the key protons could not be clearly identified.

Comment 5: 4) *The mechanistic study should be fully included in the manuscript for the sake of the readership.*

Response: Thanks for this valuable suggestion. We have added the mechanistic studies of the γ -selective silylation of allyl boronates and relevant discussions in the revised manuscript. All mechanistic studies have been included in the revised manuscript.

Comment 6: 5) *regarding compounds description: 54: the dr is not mentioned; 24 & 60 the stereogenic center of proline and ibuprofen is not mentioned.*

Response: Thanks for this suggestion. The *dr* of compound **54** is 1:1, we have revised the manuscript and Supplementary Information accordingly. Racemic starting materials of **24** and **60** were used. Therefore, the corresponding products are also racemic. This information has been added to the Supplementary Information.

Response to Reviewer # 2

Comment 1: *The manuscript by Cao, Wu, and co-workers describes a range of photoredox-catalysed hydrosilylations of alkene-containing boronic esters to synthesise borosilane products. This work builds on the authors' previously reported photoredox/HAT-catalysed alkene hydrosilylations (reference 23) by extending the reactions to alkenyl and allyl boronic esters. In contrast to their earlier work, the results for alkenyl boronic esters show unexpected regioselectivity, with the silyl radicals adding alpha to the boron centre to give geminal borosilanes. In addition, for allyl boronic esters, silyl radical addition is followed by a 1,2-boron shift to give vicinal borosilanes, thus demonstrating the versatility of the reaction conditions. Although they have reported the same regioselective hydrosilylations of alkenyl boronic esters previously with a different photoredox catalyst (reference 20), the present work provides a significantly more detailed study. The substrate scope is extensive for both the alkenyl boronic esters and the silanes, with high yields and regioselectivities obtained in most cases. The scope and limitations of the allyl boronic esters is also well*

described. The sequential reactions and derivatizations shown in Figure 2 are a nice demonstration of the synthetic utility of the chemistry, as are the scale up reactions. Finally, the authors present a detailed mechanistic study of the catalytic cycle and the regioselectivity, with DFT calculations providing a convincing explanation for the observed selectivity.

Based on the interesting regioselectivity and synthetic utility of this chemistry, I recommend publication in Nature Communications after addressing the following points.

Response: We sincerely thank Reviewer #2 for the supportive comments and valuable suggestions.

Comment 2: In Figure 3, the arrow showing the direct transformation of N to C in the mechanism is confusing, since it should proceed via HAT and then SET rather than simple deprotonation. This should be modified (or removed like in Supplementary Figure 18).

Response: Thanks for this suggestion. We have removed this arrow from N to C.

Comment 3: In Figure 3, "PCH•–" should be "PC•–" because the proton remains on I.

Response: We are sorry for this mistake. It is corrected now.

Comment 4: "Both light on/off experiments and the calculated quantum yields ($\Phi = 0.109$) supported photocatalytic cycles instead of radical chain propagation." – The mechanism in Fig. 3 is a radical chain because N and I are interconverted by sequential HAT reactions. The mechanism should be redrawn to make this clearer. As drawn, the photocatalytic cycle should be non-productive because C is converted to I and then I back to C (back electron transfer), therefore hydrosilylation should only proceed if some of I can initiate a chain. Based on the energies in Supplementary figure 20 the catalytic cycle looks energetically disfavored. Mainly because intermediates G+I undergo back electron transfer with a barrier of 1.02 kcal/mol, whereas HAT with the silane has a much higher barrier of 7.63 kcal/mol. This should be discussed. Perhaps it indicates that a radical chain propagation must be occurring to outcompete unproductive back electron transfer. The competing unproductive back electron transfer could also account for the low quantum yield because it would result in an inefficient initiation process.

Response: We agree with review 2 that a radical chain process should be at play, and have revised the mechanism in Fig. 3 and the discussion accordingly.

Revised mechanism depiction: "A plausible reaction mechanism was then proposed. The excited 4CzIPN ($E_{1/2}(PC^*/PC^{\bullet-}) = +1.35$ V vs SCE) oxidizes the thiol in the presence of DIPEA. Cyclic voltammetry measurements and DFT calculations suggest the formation of a thiyl radical species I. The thiyl radical I abstracts a hydrogen atom from silane to afford the silyl radical and thiol N. Computational studies suggested complexation with DIPEA stabilizes intermediate I and facilitates hydrogen atom abstraction from phenylsilane (Supplementary Figs. 20 and 21). Subsequently, the silyl radical adds to the α -position of the alkenyl boronate to deliver an alkyl radical intermediate which undergoes polarity-matched HAT process with thiol N to give the geminal borosilane product. The HAT steps in the proposed mechanism are supported by deuterium-labeling experiments (Supplementary Figs. 9-14). At this stage, the thiyl radical I could oxidize the reduced photocatalyst 4CzIPN•– to close the photocatalytic cycle. Alternatively, thiyl radical I could react further with silane via a radical chain pathway. However, the light on/off experiments and the calculated quantum yields ($\Phi = 0.109$) did not support an efficient radical chain process (Supplementary Figs. 15-17). To shed some light on the mechanistic pathway, computational analysis was conducted. It was

found that the back electron transfer from the reduced photocatalyst 4CzIPN^{•-} to thiyl radical **I** has an energy barrier of only 1.02 kcal/mol, while HAT from silane to thiyl radical **I** has a much higher energy barrier of 7.63 kcal/mol (Supplementary Figs. 20 -21). We attribute the ineffective chain propagation and low quantum yield to the unproductive back electron transfer.”

Revised Figure 3:

Comment 5: “¹H NMR spectroscopy showed the formation of complex **C** (Fig. 3) upon mixing ethyl thioglycolate (**A**) with equimolar DIPEA (**B**) at room temperature (Supplementary Fig. 7).” – The NMR studies for the formation of a DIPEA/thiol complex are not convincing. The chemical shifts have not changed, only the exchangeable proton on the thiol is no longer visible, therefore this is not proof for complexation. Complex formation should result in changes in the chemical shifts of the protons on both DIPEA and the thiol. Furthermore, the DFT results in Supplementary Figure 20 do not suggest that complexation should be seen by NMR, since it is 3.5 kcal/mol higher in energy.

The NMR studies should be performed in a solvent that is relevant to the reaction, such as THF-*d*8. Alternatively, an entry for CHCl₃ (or CDCl₃) should be added to Supplementary Table 2 to show that the chemistry still works in this solvent.

Cyclic voltammetry could potentially be used to confirm the formation of the DIPEA/thiol complex. However, this would require separate CVs for the thiol, DIPEA, and the thiol/DIPEA mixture. Also, these should be performed in MTBE, since the reaction fails in MeCN.

Response: Thanks for these important comments. We have performed the NMR studies in THF-*d*8, and the results indicated proton transfer between thiol and DIPEA (Supplementary Fig. 6 in the updated Supplementary Information). This is consistent with pK_a values of ethyl thioglycolate (pK_a 7.95 in water) and

DIPEA (pK_a of its conjugate acid 11.4 in water). We agree that very little change of chemical shifts is not proof of complexation.

We then turned our attention to the CV (cyclic voltammetry) experiments. Our attempt to performed CV experiments in MTBE failed due to the low solubility of supporting electrolyte in MTBE. Considering that the hydrosilylation reaction actually can be performed in MeCN for some substrates (e.g. **51** and **52**), MeCN was used for the cyclic voltammetry analysis. The results (see the figures below) showed that the CV performance of the thiol/DIPEA mixture is completely different from that of DIPEA, thiol or thiolate. In particular, a new reduction peak appeared for the thiol/DIPEA mixture, which suggested formation of a new species. Since our calculation results suggest that the complexation of thiyl radical with DIEPA (species **I** in Fig. 3) is beneficial for the reaction process, we speculate that the new reduction peak is associated with species **I**. The complexation probably occurs after the oxidation of thiol.

To summarize, the CV and calculation results suggested that the complexation of DIPEA and thiol species might occur during the reaction process, giving rise to thiyl radical species **I**. These results and discussions have been added to the revised manuscript and Supplementary Information.

$$E_{p/2}(\text{DIPEA}) = + 0.63 \text{ vs. SCE}$$

$$E_{p/2}(\text{ethyl thioglycolate}) = + 0.75 \text{ vs SCE}$$

$$E_{p/2}(\text{sodium thiolate}) = - 0.74 \text{ V vs SCE}$$

$$E_{p/2}(\text{thiol/DIPEA mixture}) = + 0.68 \text{ V vs SCE}$$

Comment 6: “Stern–Volmer fluorescence quenching and cyclic voltammetry studies indicated that the excited photocatalyst was reductively quenched by the thiol-DIPEA complex C” – Quenching studies show the plot of DIPEA quenching has a larger gradient (87.7) than that of the DIPEA/thiol mixture (81.3), implying that DIPEA is the predominant reductive quencher. This should be accounted for in the mechanistic discussion.

Response: Thanks for this comment. Stern–Volmer fluorescence quenching studies indicated that the excited photocatalyst can be reductively quenched by the mixture of thiol and DIPEA. Based on the quenching studies (see figure below), DIPEA is similarly effective as the thiol-DIPEA mixture as a quencher. We cannot rule out other mechanistic pathways. For instance, the excited 4CzIPN might oxidize the DIPEA to afford an amine radical cation species, which selectively abstracts a hydrogen atom from Si-H bond to deliver the silyl radical. Subsequently, the silyl radical adds to the α -position of the alkenyl boronate to deliver an alkyl radical intermediate which undergoes polarity-matched HAT process with thiol to give the thiyl radical and the borosilane product. The thiyl radical could oxidize the reduced photocatalyst to close the photocatalytic cycle or engage in radical chain processes. We have added these discussions to the Supplementary Information.

Comment 7: “The conformation of thiyl radical I and thiol N was suggested by DFT calculations (Supplementary Figs. 20 and 21)” – what is the relevance of this? It is not clear from the discussion in the Supplementary Information.

Response: Thanks for this comment. We have removed this discussion from the manuscript.

Comment 8: Significant emphasis has been placed on the importance of the complex between the thiol and DIPEA, despite the reaction proceeding with only slightly reduced efficiency in the absence of DIPEA. If sufficient evidence for its formation cannot be provided, the mechanism and discussion should be modified accordingly.

Response: Thanks for this comment. When the reaction time is set to 12 h, the yield of the standard product dropped from 92% to 71% without DIPEA. However, when the reaction time is set to 4 h, the yield of the standard product dropped from 92% to 54% without DIPEA. The use of DIPEA accelerates the reaction and is important for more challenging substrates. We have added these data to the manuscript and SI.

We have presented CV and calculation evidence for complex formation during the reaction process. The

mechanism and discussion have also been modified accordingly.

Comment 9: *“For unsubstituted vinyl boronate, a 1 to 1.7 ratio of α -silylation and β -silylation products (28) was noted, possibly due to the steric effect during the radical addition” – could this also be a result of a 1,2-boron shift of the primary alkyl radical intermediate?*

Response: Thanks for this comment. A 1,2-boron shift of the primary alkyl radical intermediate might also be possible. We have removed the sentence “possibly due to the steric effect during the radical addition”.

Comment 10: *Page S37 discusses how to “determine the photon flux of the spectrophotometer”. This should be the photon flux of the Kessil lamp used to measure the quantum yield. The method for determining the photon flux states 3.0 mL of ferrioxalate solution was used. But later it states VI = 2.0 mL and 2.0 mL of MTBE was used in the reaction to measure the quantum yield. These numbers need to be checked and corrected.*

Response: Thanks for this comment. We are sorry for the incorrect writing and have revised the Supplementary Information accordingly. 2.0 mL of ferrioxalate solution was used.

Comment 11: *In Supplementary Figure 19, the gamma symbols have been replaced by “?”.*

Response: We are sorry for the mistake and have revised accordingly.

Response to Reviewer # 3

Comment 1: *The authors in this manuscript report synthesis of geminal and vicinal borosilanes via photocatalyzed regioselective hydrosilylation. The reported synthetic protocol looks interesting. Detailed mechanistic studies were also performed. Since I am not a synthetic chemist, I will let colleagues in the synthetic area comment on the importance of the reactions reported. As requested by the editor, I will mainly comment on the DFT calculation part.*

Response: We sincerely thank reviewer #3 for the comments and valuable suggestions.

Comment 2: *The DFT results are clearly consistent with the experimental findings regarding the regioselectivity. While the authors employed the steric argument to rationalize γ -selective silylation of allyl boronates, it is unclear why the steric effect becomes unimportant in α -selective silylation of alkenyl boronates. The authors owe readers a clear picture regarding what they have said on the selectivity issues. In the silylation of alkenyl boronates, radical addition intermediate T2 (Figure 4) is more stable than T1. However, the barrier leading to T2 is higher. An explanation is needed for readers to understand the DFT results better.*

Response: Thanks for these valuable comments. We have performed DFT calculation regarding β - or γ -selectivity for allyl boronate (see the figure below or Supplementary Fig. 24). The energy barrier of silyl radical adding to the γ -position of **R**³ is 1.83 kcal·mol⁻¹ lower than that to β -position (**S**⁵ vs **S**⁶). Moreover, the energy barrier for HAT from **N** to the γ -selective radical adducts **T**⁵ or **T**⁷ is lower than that to the β -

selective radical adduct T^6 . Therefore, γ -selective radical addition is favored for the allyl boronate.

For alkenyl boronate, the calculated energy diagram illustrates that the addition of a silyl radical to alkenyl boronic esters determines the regioselectivity because the transition states (S^1 or S^2) have the highest energy in the reaction pathways. The energy barrier of silyl radical adding to α -position of R^1 is $1.64 \text{ kcal}\cdot\text{mol}^{-1}$ lower than that to the β -position (S^1 vs S^2), which means the α -addition rate is approximately 16 times faster than the β -addition. The radical addition processes are nearly irreversible at room temperature, thus the equilibrium between α - and β -addition products cannot be reached.

We have added more discussions to the manuscript and SI to make it clear.

Revised supplementary figure 24:

Comment 3: Since this is a manuscript submitted to Nature Communications, I expected that the authors should explain why they use M06-2X functional instead of others in their calculations. I think the authors should also use some other functionals to examine if different functionals give consistent results.

Response: Thanks for this suggestion. It's well-known that the functional M06-2X is suitable for calculating the reaction energy and barrier for organic or main-group element systems. Many previous benchmark studies have demonstrated its accuracy and robustness (*Phys. Chem. Chem. Phys.*, **19**, 32184–32215

(2017)). To examine whether different functionals give consistent results, we used 4 different types of functionals with great performance in benchmark studies (MN15-D3(BJ), PW6B95-D3, B3LYP-D3(BJ) and wB97XD) to study the α -selectivity in the silylation of alkenyl boronates. Consistent results were obtained with very similar $\Delta\Delta G$ values (see the table below). These data have been added to the revised Supplementary Information.

	S ¹	S ²	$\Delta\Delta G(S^1-S^2)$	T ¹	T ²	$\Delta\Delta G(T^1-T^2)$
M06-2X-D3	6.76	8.40	-1.64	-11.03	-12.82	1.79
MN15-D3(BJ)	5.81	6.83	-1.02	-12.15	-14.33	2.18
PW6B95-D3	6.22	8.14	-1.92	-9.83	-11.57	1.74
B3LYP-D3(BJ)	5.05	7.57	-2.52	-9.18	-10.96	1.78
wB97XD	6.57	8.97	-2.40	13.30	-14.47	1.17

	U ¹	U ²	$\Delta\Delta G(U^1-U^2)$	V ¹	V ²	$\Delta\Delta G(V^1-V^2)$
M06-2X-D3	0.68	1.79	-1.11	-18.92	-16.57	-2.35
MN15-D3(BJ)	1.18	2.50	-1.32	-18.34	-16.24	-2.10
PW6B95-D3	0.74	2.53	-1.79	-16.70	-14.08	-2.62
B3LYP-D3(BJ)	-0.35	0.95	-1.30	-16.19	-13.72	-2.47
wB97XD	-3.60	-1.62	-1.98	-20.85	-18.26	-2.59

Comment 4: *The DFT results provided in the Supplementary Information are poorly organized and described.*

(i) First, I expect that the energy profiles given in Fig. S20, 22 and 23 should have the same quality as the one presented in Fig. 4 in the main text. Readers easily comprehend the energy profiles given in the main text Fig. 4. However, it is very hard to understand and comprehend the energy profiles given in the SI.

(ii) The description and discussion given in the SI are also far poorer than those given in the main text. The English should be polished with significant effort.

(iii) Fig. S20 should be completely redrawn. I do not understand the symbol placed between G+M+N and G+I. In the related discussion, the authors should clearly spell out the most favorable pathway. Ideally, the authors provide a sketch of the favorable cycle in the SI.

Response: Thanks for this suggestion. We have redrawn the relevant figures to make them clear. The description and discussions have also been revised.

Reviewers' Comments:

Reviewer #1:

Remarks to the Author:

This manuscript from Cao and Wu and co-workers describes the divergent hydrosilylation of vinyl boron to access either geminal or vicinal borylated/silylated carbon center.

I already reviewed this manuscript and I had a careful look at the revised version from the authors. Some points highlighted by the other reviewers were addressed. However, I am still not convinced of the interest of this work for the readership of the journal.

In addition, I am unhappy with the correction made by the authors regarding my own comments:

1/ I have some concerns with the claim from the authors that the addition on vinyl borane is beta-selective, example from the Leonori lab (ACS Catalysis) demonstrated and rationalized the alpha addition on this backbone;

2/ The literature is still not well cited, the work from Tanaka and Poisson deserve to be cited, as they reported previous photoinduced and electrochemical hydrosilylation reaction. In light of this report, the novelty of this work is highly questionable;

Hence, I cannot recommend this work for publication in Nature Communications in terms of novelty and urgency. As I already mentioned, this manuscript deserves publication in a more specialized journal focusing on either organic chemistry or catalysis.

Reviewer #2:

Remarks to the Author:

All my concerns have been fully addressed in the revised manuscript and supporting information.

Reviewer #3:

Remarks to the Author:

In this revised version, the authors have satisfactorily addressed my comments given in the previous round of review.

In other words, the revised version is now fine with me.

Response to Reviewer # 1

Comment 1: *This manuscript from Cao and Wu and co-workers describes the divergent hydrosilylation of vinyl boron to access either geminal or vicinal borylated/silylated carbon center.*

I already reviewed this manuscript and I had a careful look at the revised version from the authors. Some points highlighted by the other reviewers were addressed. However, I am still not convinced of the interest of this work for the readership of the journal.

In addition, I am unhappy with the correction made by the authors regarding my own comments:

I have some concerns with the claim from the authors that the addition on vinyl borane is beta-selective, example from the Leonori lab (ACS Catalysis) demonstrated and rationalized the alpha addition on this backbone;

Response:

Thanks for this important comment. The addition of carbon-centered radicals to alkenyl boronic esters have been reported to be beta-selective because the formed α -boryl radicals could be stabilized by delocalization to the empty p-orbitals of the adjacent boron atoms (ref. 33-38). This type of radical addition process has been reviewed by Morken and co-workers (*Eur. J. Org. Chem.* 2020, 2362–2368, ref. 38). Therefore, it is quite surprising that alfa-selective silylation of alkenyl boronic esters was observed in our study.

In Leonori's study (*ACS Catal.* 2017, 7, 4126–4130), alkenyl potassium trifluoroborates were used as the radical acceptors. The boron atom in potassium trifluoroborate is four-coordinated and lacks any empty p-orbital, thus providing no stabilization effect to an alfa-carbon radical (*J. Am. Chem. Soc.* 2015, 137, 6762–6765). The authors attributed the observed alfa-selectivity to higher spin density and more favorable reaction enthalpy.

Because boronates include both esters and salts of boronic acids, we have changed “boronates” to “boronic esters” in the introduction. We apologize for the inaccurate description.

Comment 2: *The literature is still not well cited, the work from Tanaka and Poisson deserve to be cited, as they reported previous photoinduced and electrochemical hydrosilylation reaction. In light of this report, the novelty of this work is highly questionable.*

Hence, I cannot recommend this work for publication in Nature Communications in terms of novelty and urgency. As I already mentioned, this manuscript deserves publication in a more specialized journal focusing on either organic chemistry or catalysis.

Response:

Thanks for this suggestion. We have added the hydrosilylation work from Tanaka and Poisson to references (ref. 27,28).

We respectfully disagree with the comment “*In light of this report, the novelty of this work is highly questionable*”. As we have pointed out in the response to your comment 1, the novelty of this work is the unusual regioselectivity in radical hydrosilylation and detailed mechanistic insights. This study provides evidence that radical approaches could be used for challenging regiocontrol, and will help pushing the boundaries of selective radical functionalization of alkenes. Other important characters of this work include: 1) a rare example of α -selective radical functionalization of alkenyl boronates because the addition of radicals to alkenyl boronates has been known to be β -selective; 2) the first general strategy to synthesize geminal and

vicinal borosilanes via radical pathways; 3) green and sustainable synthesis of borosilanes; 4) access to diverse multi-borosilanes which serve as attractive building blocks for material science.

Response to Reviewer # 2

Comment 1: All my concerns have been fully addressed in the revised manuscript and supporting information.

Response: We sincerely thank Reviewer #2 for the supportive comments and valuable suggestions.

Response to Reviewer # 3

Comment 1: In this revised version, the authors have satisfactorily addressed my comments given in the previous round of review. In other words, the revised version is now fine with me.

Response: We sincerely thank Reviewer #3 for the supportive comments and valuable suggestions.